# Bacteria–Cancer Interface: Awaiting the Perfect Storm

**DOI:** 10.3390/pathogens10101321

**Published:** 2021-10-14

**Authors:** Jonathan Pommer Hansen, Waled Mohammed Ali, Rajeeve Sivadasan, Karthika Rajeeve

**Affiliations:** 1Department of Biomedicine, The Skou Building, Høegh-Guldbergs Gade 10, 8000 Aarhus, Denmark; 201709757@post.au.dk (J.P.H.); wili.ali94@gmail.com (W.M.A.); 2Neurobiology Laboratory, Rajiv Gandhi Centre for Biotechnology, Thiruvananthapuram, Kerala 695014, India; rajshiv@rgcb.res.in; 3Pathogen Biology Laboratory, Rajiv Gandhi Centre for Biotechnology, Thiruvananthapuram, Kerala 695014, India

**Keywords:** bacteria, cancer hallmarks, inflammation, DNA damage, epithelial mesenchymal transition

## Abstract

Epidemiological evidence reveal a very close association of malignancies with chronic inflammation as a result of persistent bacterial infection. Recently, more studies have provided experimental evidence for an etiological role of bacterial factors disposing infected tissue towards carcinoma. When healthy cells accumulate genomic insults resulting in DNA damage, they may sustain proliferative signalling, resist apoptotic signals, evade growth suppressors, enable replicative immortality, and induce angiogenesis, thus boosting active invasion and metastasis. Moreover, these cells must be able to deregulate cellular energetics and have the ability to evade immune destruction. How bacterial infection leads to mutations and enriches a tumour-promoting inflammatory response or micro-environment is still not clear. In this review we showcase well-studied bacteria and their virulence factors that are tightly associated with carcinoma and the various mechanisms and pathways that could have carcinogenic properties.

## 1. Introduction

Genetic alterations that are inherited or acquired during a person’s lifetime can lead to errors in cell division and uncontrolled growth. External factors leading to these genetic mutations largely include exposure to radiation, smoking, and infectious microorganisms such as viruses, bacteria, and parasites. Data from the Global Cancer Observatory (GLOBOCAN) attribute 2.2 million (13%) new cancer cases to 10 carcinogenic pathogens. The International Agency for Research on Cancer (IARC) has classified these carcinogens under six viruses (Epstein—Barr virus, human papillomavirus, hepatitis virus B, hepatitis virus C, human herpesvirus type 8 and human T cell lymphotropic virus type 1), one bacterium (*Helicobacter pylori)* and three parasites (*Opisthorchis viverrine*, *Clonorchis sinensis*, and *Schistosoma haematobium*). These classifications do not include the pathogens such as the human immunodeficiency virus that results in immunosuppression and enhances the carcinogenic action of viruses and probably bacteria. While only one bacterium has so far been included to the IARC’s list of carcinogenic pathogens, many other bacteria have been discovered to have carcinogenic effect. In this review we will shed light into all the known cancer-associated bacterium (Table 1, Figure 1) and discuss the factors and pathways that the pathogens activate to boost malignancies. The bacterial infection can contribute to carcinoma by favouring inflammatory processes and the release of carcinogenic bacterial effectors.

### 1.1. Helicobacter pylori

*H. pylori* is a Gram-negative bacterium, that has established lifelong colonization in the stomach of two-thirds of the world’s population with the highest prevalence in Africa [35]. According to the GLOBOCAN database, *H. pylori* is responsible for 36.8% of gastric cancers out of the estimated 2.2 million cancers attributed to infections [36]. From the 850,000 non-cardia gastric cancer cases worldwide in 2018, 89% of the total incidence are attributable to *H. pylori* infections, whereas around 73% of the non-Hodgkin lymphoma of gastric location are attributable to *H. pylori* infections [36]. Furthermore, *H. pylori* is the only bacterium classified as a group 1 carcinogen by IARC. 

As a result of both intra and intergenomic diversification, *H. pylori* strains have shown to appear with a broad genetic diversity [37]. Certain strain-specific proteins such as vacuolating cytotoxin A (VacA) and cytotoxin-associated gene A (CagA) are shown to have increased risk towards carcinoma [38,39,40,41]. VacA is a channel forming toxin capable of inducing vacuoles into epithelial cells and has long been associated with the development of gastric inflammation [38]. While the exact oncogenic mechanisms have not yet been established, potential pathways have been suggested: VacA was found to cause apoptosis in gastro-epithelial cells leading to increased cell proliferation or a possible entry point for other carcinogens into the gastric mucosa [38]. VacA can activate the phosphoinositide 3-kinase (PI3K/AKT) pathway, resulting in the inhibition of glycogen synthase kinase 3 beta (GSK3β) via phosphorylation, with the subsequent dissociation of the GSK3β/β-catenin complex [42].

CagA is shown to have a direct oncogenic effect and its expression can lead to atrophic gastritis and gastric cancer [43]. The bacterium uses type IV secretion system (T4SS) to inject CagA into gastric epithelial cells; the protein undergoes tyrosine phosphorylation by the proto-oncogene tyrosine protein kinase (SRC) at the EPIYA phosphorylation motif (Glu-Pro-Ile-Tyr-Ala amino acid sequence). Phosphorylated CagA activates SHP-2 (SRC homology region 2 domain-containing phosphatase-2), a tyrosine phosphatase and known oncoprotein, which potentially activates β-catenin, Janus kinases/signal transducer and activator of transcription proteins (JAK/STAT) [44] and nuclear factor like kappa B (NF-kB) [45] pathways resulting in targeted transcriptional upregulation of genes involved in carcinogenesis [46]. c-Src inactivation induced by CagA leads to tyrosine dephosphorylation of an actin binding protein, cortactin, thus rearranging the actin cytoskeleton [47,48]. Cortactin expression promotes the migration, invasion, and proliferation of cells in vitro and in vivo, conferring malignancy [49,50]. Recently, *Helicobacter* was reported to upregulate cortactin in a CagA and JNK-dependent pathway which might partially explain the mechanism of *Helicobacter* driven carcinogenic process [51] (Figure 2). 

CagA activates the extracellular signal-regulated kinases/mitogen-activated protein kinase (ERK/MAPK) pathway through SHP2, resulting in morphological changes in the epithelium called hummingbird phenotype in a Ras-independent manner [52]. CagA also interacts with the E-cadherin/β-catenin complex resulting in nuclear accumulation of β-catenin leading to transdifferentiation of gastric epithelial cells [53]. The bacteria can induce the epithelial-to-mesenchymal transition (EMT) mediated by Snail, a transcriptional repressor of E-Cadherin expression, in gastric epithelial cells by GSK3β depletion [54]. The tumour suppressor TP53 undergoes proteasomal degradation leading to DNA damage in the presence of CagA [55]. Most interestingly, the transgenic expression of CagA in mice induces gastrointestinal and hematopoietic neoplasms, revealing the direct carcinogenic potential of the protein [41]. 

Other proteins that are associated with *H. pylori,* which increases the risk of gastric cancer, are the outer membrane proteins (OMPs) such as blood-group antigen-binding adhesin (BabA) and outer inflammatory protein A (OipA). BabA binds to human Lewis(b) surface epitopes, which results in increased levels of specific cytokines causing cell proliferation. OipA activates β-catenin through binding of epidermal growth factor receptor and activation of PI3K-AKT signalling [56,57] (Figure 2). The eradication of *H. pylori*, in both healthy individuals and individuals with gastric neoplasia, reduces the incidence of gastric cancer and hence the mortality from gastric cancer [58].

### 1.2. Salmonella enterica and Salmonella typhi 

*Salmonella*, a Gram-negative bacterium from the Enterobacteriaceae family of which subspecies *Salmonella Enteritidis* (*S. enteritidis*) and *Salmonella typhi* (*S. typhi*) have been associated with an increased risk of colon cancer [63] and gallbladder carcinoma, respectively [10,11,64]. *Salmonella enterica* infection is known to modulate inflammatory response of the host leading to DNA damage, increased cell proliferation, and migration leading to neoplasm and eventually cancer [65,66]. *Salmonella* effector proteins have been shown to be associated with colon cancer. The first effector protein is typhoid toxin, a cyclomodulin, that favours dysbiosis and leads to increased cell survival, risking the development of inflammatory bowel disease and colon cancer [67,68]. The second effector protein is Avr, secreted by T3SS and detected in the stool of patients with colon cancer [69]. Avr decreases inflammation as it suppresses secretion of cytokines such as IL-12, IL-6, IL-20, TNF-α, and interferon-γ. Avr is shown to directly activate the Wnt/catenin pathway favouring tumour formation in intestinal epithelium [70]. The acetyl transferase activity of Avr targets p53 [71], leading to cell cycle arrest and apoptosis inhibition by depleting the pro-apoptotic protein Bax [72]. Finally, the effector protein AvrA was found to increase the STAT3 signalling pathway in colon cancer [73]. STAT3 is a transcription factor, known to increase cell proliferation and inflammatory regulation and promote tumorigenesis. 

*Salmonella* sp. enters the gall bladder through the bloodstream or the bile [74]. Several studies have shown the presence of *Salmonella enterica*, *S. typhi*, *S. paratyphi*, *S. typhimurium* in tissue biopsies of gall bladder carcinoma [75]. *Salmonella* has the ability to form biofilms on gall stones and can persist in the infected tissue [76]. The inflammation caused during infection may cause immune cell recruitment including macrophages that express COX-2, an enzyme that is crucial in the development of gastro-intestinal tumors [77,78]. A study done on murine gallbladder organoids reports that *Salmonella enterica* was not a single causal factor, but a part of a multistep process in the development of gallbladder carcinoma [10]. Overexpression of the proto-oncogene c-Myc and silencing of tumour suppressor p53, together with the presence of the MAPK and AKT pathways, was necessary to provoke cellular transformation in the organoids [10]. *Salmonella enterica* was shown to activate MAPK and AKT pathways through effector proteins such as SopB, SopE and SopE2, injected by T3SS into host cells to regulate bacterial uptake and secure intracellular survival through cell transformation [10] (Figure 3). However, more studies are required to study which patients are predisposed to *Salmonella*-induced carcinoma. 

### 1.3. Escherichia coli 

*Escherichia coli* (*E. coli*) is a Gram-negative, facultative anaerobe commensal bacteria found in the gut. Studies have frequently found strong associations between mucosa-adherent *E. coli* strains and colorectal cancer (CRC) [79]. Genotoxins are produced by bacteria as a means of competing with other microbes. The cyclomodulin-positive B2 *E. coli* strain possesses the polyketide synthase genetic islands (*pks* +ve), coding for the bacterial genotoxin colibactin. This *E. coli* strain abnormally colonizes the mucosa of colon cancer patients, promoting low grade inflammation and cell proliferation [80]. Colibactin has shown carcinogenic properties by inducing double-stranded DNA breaks and cell cycle arrest [81]. This activates G2-M DNA damage checkpoint pathway where the mismatch repair proteins MSH2 and MLH1 are effectively depleted in colonic cells through T3SS-induced effector proteins [81]. Colibactin was shown to promote colon tumour growth by induing senescence-associated secretory phenotype through increased p53 SUMOylation [82]. In vitro studies show that the bacteria can alter tumour microenvironment by inducing the three major hallmarks of cancer: EMT, altering the cancer stem cell population, and metabolic reprogramming [83]. Most interestingly the etiological role of colibactin in colorectal carcinoma was evident when the DNA damage signature induced by colibactin corresponded to mutational hotspots in the genome of colorectal cancer patients [5,84]. Nevertheless, whether an already existing colon cancer creates a favourable environment for the B2 *E. coli* to colonize the colonic epithelial cells [82], or whether *E. coli* is a causal agent for CRC is still unclear (Figure 4).

### 1.4. Chlamydia trachomatis and pneumoniae 

The Chlamydiaceae family is a group of obligate intracellular bacteria, with the subspecies *Chlamydia trachomatis* (*C. trachomatis)* capable of causing sexually transmitted infections and *Chlamydia pneumoniae (C. pneumoniae)* capable of causing lung infections. Both these bacteria cause asymptomatic and chronic infections. The first associations between *C. pneumoniae* and lung cancer goes far back, however, newer studies have found varying results when trying to establish a reliable biomarker in measuring chronic *C. pneumoniae* infection and linking it to lung cancer [85,86]. Most often the measure of titre immunoglobulin A was used, in which some studies found associations, while others found none. Newer studies found associations by measuring *Chlamydia* heat shock protein-60 antigen, another marker used to indicate chronic *Chlamydia* infection. The conflicting results could be caused by the modest reliability of micro-immunofluorescence. Despite controversial results, most studies suggest that chronic *C. pneumoniae* infection did associate with lung cancer [87,88]. Recent studies have indicated that the association may not only be mediated by inflammation, but also epigenetic changes, such as transformation of mesothelial cells and dysregulation in replication, transcription, and DNA repair mechanisms in the host cell [89]. How *C. pneumoniae* induces these epigenetic changes is unclear and should be looked further into. 

While *human papillomavirus* (HPV) is well known as the most common cause of cervical cancer inflicted by infections, studies indicate that *C. trachomatis* may not just be a comorbidity to HPV, but instead an independent predictor for cervical cancer. The first evidence of *C. trachomatis* associated to cancer was based on an epidemiological correlation between *C. trachomatis* reactive serum antibodies and the presence of cervical cancer [90,91,92]. Studies show the presence of chlamydia HSP60-1 antibodies associated with increased cervical cancer risk [93]. Recently, studies from two independent population showed women seropositive for *C. trachomatis* had twice the risk of developing ovarian carcinoma [94]. The mechanism by which *C. trachomatis* may lead to cancer development is described through its ability to cause the host cell DNA damage and an impairment of the host DNA damage response [30]. Increased levels of reactive oxygen species (ROS) [95] lead to oxidative DNA damage, resulting in the production of single-strand breaks [96]. The impairment of the host DNA damage response was caused by histone alterations and activation of other repair inhibitory cell mechanisms. Through inhibitions of base excision repairs, DNA damage was not repaired efficiently, resulting in genomic instability. *C. trachomatis* is known to strongly activate the PI3/MAPK survival signal and efficiently block cell death [97,98]. It also activates the anti-apoptotic protein cIAP-2. The bacterium is known to possess deubiquitinase Cdu1, which is known to stabilize the anti-apoptotic protein, Mcl-1 [99,100]. Moreover, *C. trachomatis* also downregulates the tumour suppressor p53 [27,31] via the PI3K and MDM2 pathways. The bacteria induce glutamine addiction in the infected cells by stabilizing the proto-oncogene c-Myc [26]. In addition, the infection remains asymptomatic by paralyzing the neutrophils, the first line of defense [32]. Persistent infection of *C. trachomatis*-inducing DNA damage, blocking cell death and preventing immune surveillance might lead to tissue damage and malignancies in the infected tissue (Figure 5). 

### 1.5. Fusobacterium nucleatum 

*F. nucleatum* is a Gram-negative obligate anaerobe bacterium commonly found in the oral cavity, known to cause opportunistic infections [101]. This oral commensal is a periodontal pathogen associated with a wide spectrum of human diseases. *F. nucleatum* has been associated with CRC through detection by quantitative polymerase chain reaction and 16S rRNA sequence [102]. Recently, *F. nucleatum* has been associated with additional cancer types, including oral, head and neck, oesophageal, cervical, and gastric cancers [103]. However, additional studies should be conducted to further back these associations. 

The role of *F. nucleatum* in CRC is thought to be through multiple different pathways. Fusobacterium adhesin A (FadA) was found to bind and inhibit E-cadherin in xenograft mice. E-cadherin is a tumour suppressor, causing activation of β-catenin signalling, previously mentioned to promote cell proliferation [104]. Additionally, FadA increased expression of nuclear factor-κB (NF-κB) and pro-inflammatory interleukins such as IL-6, IL-8, and IL-18 leading to a proinflammatory microenvironment [104]. Finally, *F. nucleatum* was shown to induce anticancer immune response evasion through fusobacterial apoptosis protein 2 (Fap2) [105]. Fap2 was found to bind the TIGIT immune receptor, a receptor expressed on all natural killer cells (NK cells) and other immune cells, thus inhibiting NK cell cytotoxicity [105]. Additionally, TIGIT was expressed on tumour-infiltrating lymphocytes and Fap2 was found to inhibit T cell activities, effectively protecting *F. nucleatum* from immune cell attacks (Figure 6). Interestingly, it was shown that exosomes harbouring miR1246/92b-3p/27a-3p CXCL16 derived from *F. nucleatum*-infected colorectal cancer cells facilitated tumour metastasis [106]. Moreover, in vitro and in vivo studies show that the bacterial infection leads to upregulation of cytochrome p450 monooxygenase, which increases the invasiveness and migratory ability (EMT) of colorectal cancer cells via the TLR4/Keap1/NRF2 axis [107].

### 1.6. Streptococcus bovis

*S. bovis* is a group of Gram-positive, opportunistic bacteria, capable of causing endocarditis and bacteraemia. *S. bovis* has shown to associate with colorectal cancer (CRC), through a series of retrospective studies [108,109]. Over time it has become clearer, that the association was mostly derived from the subspecies *Streptococcus gallolyticus* (SG). While it was previously uncertain if the presence of SG in tumour tissue was caused by SG-promoting tumour development, or the result of the tumour enabling SG colonization, recent studies done on mice found that SG may in fact promote colorectal tumour development [110,111]. 

The study showed a clear indication that SG increased levels of β-catenin, c-Myc, and proliferative cell nuclear antigen (PCNA) resulting in cell proliferation in certain responsive cell lines. Additionally, silencing β-catenin through knockdown cells resulted in a completely abolished effect of SG on the increased cell proliferation, thereby indicating the important role of SG in activating β-catenin and promoting cell proliferation, resulting in potential tumour development. The pathway of how SG upregulated β-catenin is still unclear and should be further investigated [111] (Figure 7).

### 1.7. Mycoplasma 

*Mycoplasma* is a group of small bacteria with no cell wall, consisting of more than 100 subspecies. Several species of *Mycoplasma* have been suspected to associate with cancer, with the most prominent being: *M. hyorhinis*, *M. hominis*, and *M. genitalium*, and a few mentions of *M. penetrans* and *M. salivarium* [112,113,114]. 

Few studies have attempted to find these associations, indicating a need for additional large-scale studies. Whether these bacteria can directly lead to malignancies is yet unclear. In relation to cancer, *M. hyorhinis* is the most widely studied and has shown to be associated with prostate cancer, gastric carcinoma, oesophageal cancer, lung cancer, and breast cancer. In recent studies, several of the bacterial proteins have been found to possibly have carcinogenic mechanisms. One study found a clear indication of *M. hyorhinis* activating the β-catenin signalling pathway in *M. hyorhinis*-induced gastric cancer cell motility. Additionally, the activation of β-catenin seemed to be induced by glycogen synthase kinase 3 beta (GSK3β) and the Wnt-receptor lipoprotein-receptor-related protein 6 (LRP6), while also finding interaction between LRP6 and the mycoplasmal membrane protein p37. When GSK3β did not bind and activate LRP6, no increased activation of β-catenin was found [34]. The p37 protein activates carcinogenic effects such as enabling immortality in mammalian cell lines [115] and promoting cell motility, migration, and invasion through the activation of metalloproteinase-2 and epidermal growth factor receptor *in vitro* [116] (Figure 8).

### 1.8. Bacteroides fragilis 

*B. fragilis* is an anaerobe bacterium, part of the normal microbiota in the human colon, often categorized into nontoxigenic *B. fragilis* and enterotoxigenic *B. fragilis* (ETBF), capable of producing a heat-labile toxin [117]. Recently, gut microbiota have been shown to play a bigger role than previously anticipated in the development of CRC, and ETBF has been found to be prevalent in the colonic mucosa of CRC patients [117,118]. This data was additionally supported by a study on APC^+/−^ mice, which showed the toxin from ETBF had carcinogenic properties through inflammatory cascades. The *B. fragilis* toxin (BFT) mediated E-cadherin cleavage, through an unidentified receptor, resulting in the activation of the Wnt/β-catenin signalling pathway [119]. Additionally, BFT was able to create a proinflammatory microenvironment through activation of STAT3 and possibly NF-κB. The study additionally found that IL-17 is a key proinflammatory mediator necessary for colon tumour development in the APC^+/−^ mice [119] (Figure 9). The link between IL-17 and tumorigenesis is still unclear. 

### 1.9. Neisseria gonorrhoeae 

*N. gonorrhoeae* is an aerobic Gram-negative bacterium, well known for its diplococci shape, being one of the most frequent sexually transmitted diseases, and its capability to cause pelvic inflammatory disease. *N. gonorrhoeae* has widely been tested for its association with prostate cancer, often leading to controversial results, with large prospective cohort studies failing to find any association [120]. A more recent meta-analysis suggests that despite the varying results, *N. gonorrhoea* may in fact be associated with prostate cancer [21]. Other studies found associations between *N. gonorrhoeae* and bladder cancer in men [23]. Due to the vast amount of controversy, large-scale cohort studies should be conducted to support these associations. The mechanism behind the potential carcinogenic effect of *N. gonorrhoeae* is unclear and not well explored, likely due to its inconsistent associations. However, one study found *N. gonorrhoeae* caused DNA damage, increased the expression of p21 and p27 and decreased the expression of p53 in non-tumour epithelial cells. This could support the claim that *N. gonorrhoeae* may be carcinogenic, although this should be further explored [20] (Figure 10). 

### 1.10. Enterococcus faecalis 

*E. faecalis* is a facultative anaerobe Gram-positive bacterium commonly found in the oral cavity and intestines. *E. faecalis* populations have been found to be significantly increased in faeces of patients with CRC compared to healthy controls; however, the question of whether the role of *E. faecalis* is protective or carcinogenic remains [121]. One study found *E. faecalis* to activate Wnt/β-catenin signalling through coculturing with M1 macrophages, thereby inducing cell proliferation in murine primary colon epithelial cells [122]. Another study found the heat-killed *E. faecalis* EC-12 strain supressed β-catenin signalling, thus having a potential protective role in CRC [123]. 

A possible explanation could be caused by the isolation of different *E. faecalis* strains. Different strains, as the result of gene transfers, could lead to the bacterium becoming more or less virulent, resulting in different inflammatory responses [124]. A harmful role of *E. faecalis* has been suggested to be through its ability to induce intracellular ROS production, causing DNA instability and a lowered DNA repair response. Finally, infection seemed to induce a NF-κB-dominated inflammatory response [125] (Figure 11) 

### 1.11. Clostridium septicum 

*C. septicum* is an anaerobic, Gram-positive, gas-producing bacterium capable of forming spores. Infection with *C. septicum* is uncommon. Despite the low number of cases, studies have found strong indications of its associated with CRC [126,127]. It is not clear if this association is a coincidence or if any bacterial virulence factors are directly causing malignancy. A possible explanation of this association has been suggested to be that the tumour creates a hypoxic and acidic microenvironment through anaerobic glycolysis, which favours the germination of the *C. septicum* spores, indicating that the bacterium is not the causal agent of the tumour [128]. A direct involvement of *C. septicum* in CRC has not yet been found. 

### 1.12. Mycobacterium tuberculosis 

*Mycobacterium tuberculosis* (MBT) is an intracellular pathogenic bacterium, capable of entering the respiratory airways, establishing lifelong infections. MBT has been well documented to be associated with increased risk of lung cancer [129]. A recent study found strong associations with cancer, specifically from the *M. tuberculosis* L form (MBT-L), a wall defective and pleomorphic form of MBT, which behaves much like carcinogenic viruses. The MBT-L had low activity and pathogenicity, which may often have gone undiagnosed causing possible misclassifications such as chronic lymphadenitis of unknown origin [130]. Recently a study on mice, found MBT inhibited T-cell-mediated cellular immune response by impairing T helper cells in the late stage of infection. MBT is reported to promote tumour metastasis in the lungs through increased programmed cell death protein 1/programmed death-ligand 1 (PD-1/PD-L1) pathway activation [11] (Figure 12). 

## 2. Discussion

Although a rather large group of bacteria has been shown to be associated with different cancer types, only *H. pylori* is currently accepted as a group 1 carcinogen, reflecting how the bacterial cancer field has not received the same amount of attention as the viral field. While some bacteria such as *E. coli*, *S. bovis,* and *C. trachomatis* have long been associated with cancer, studies focused on bacterial factors that directly predispose infected tissue towards carcinoma have only recently been reported. The bacterial effectors that can inflict mutations and result in malignancies in the infected tissue are yet to be discovered. 

Several bacteria mentioned are commonly found in the human microbiome, yet they are associated with cancer. If a bacterium is causal for cancer, why is the cancer not seen more frequently? Even though cancer is one of the well-funded and focused research areas, much is still to be discovered. For a cell to turn malignant due to chronic infectious disease, the cell must accumulate the respective mutations in the DNA, or these epigenetic changes might imprint memory to activate indefinite survival signals leading to cell proliferation. Most of the showcased bacteria are capable of inflicting one or more of these hallmarks to an otherwise normal cell, but unless all the hallmarks are acquired, i.e., “the perfect storm has hit”, the cell may not be a fully functional cancer cell. If the bacteria truly are linked to the cause of cancer, they are not likely the only causal factor but instead a contribution in the multistep process required to fulfil the hallmarks of cancer, leaving the rest of the hallmarks to be acquired by other causes, such as radiation, smoking, aging, or inheritance Thus these chronic infections can increase the fitness of cells towards carcinoma. 

Since many studies show that bacteria could be located at the site of tumour development, it is also debatable whether the bacteria might colocalize in the transformed tissue due to the excess nutrient availability and increased vascularization. This affinity for tumour tissue is exploited by scientists, by using attenuated strains of bacteria as adjuvants in the treatment of various neoplasms. The bacterial tropism for tumour microenvironment could also activate the innate and adaptive immune response of the host to clear malignant cells. 

This review showcases that bacteria likely have a bigger role to play in the category of infectious cancers, and that this category, previously estimated to be 13% of all cancers, is likely underestimated. The bacteria included in this review are the most suggested bacteria to be associated with increased risk of cancer, yet there is still a lack of deeper understanding in how these bacteria may be associated to the cause of cancer. This highlights the need for more research to be done in this field with the likelihood of finding more bacteria, not mentioned in this review, to also have carcinogenic properties. 

## Figures and Tables

**Figure 1 pathogens-10-01321-f001:**
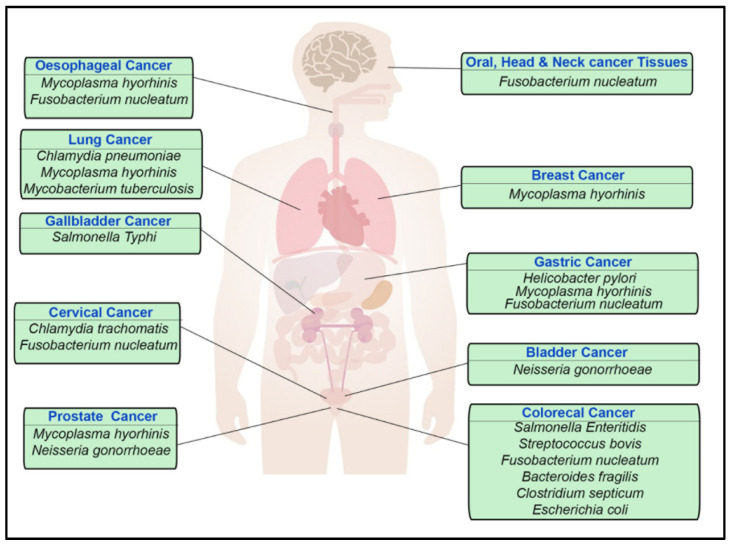
Graphical overview of bacteria associated with carcinoma.

**Figure 2 pathogens-10-01321-f002:**
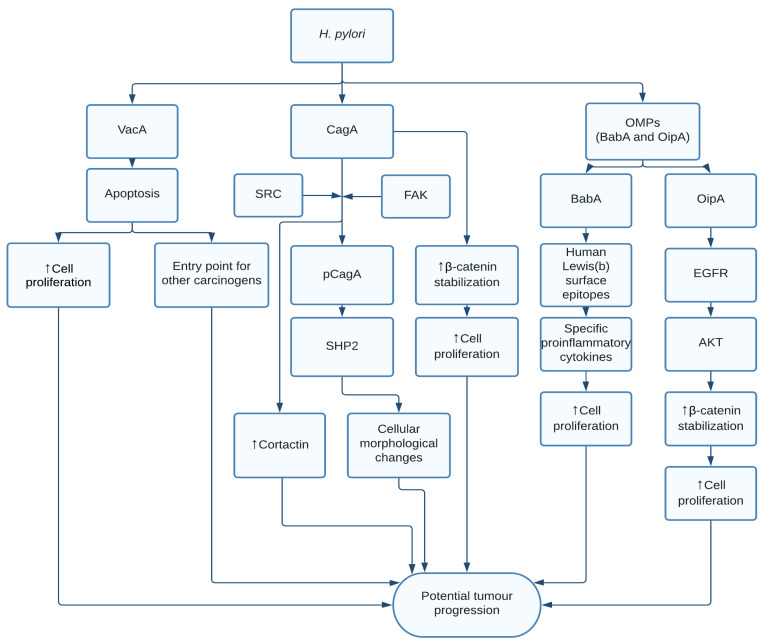
Oncogenic pathways activated by *H. pylori:* VacA can cause apoptosis, resulting in increased cell proliferation or serve as a possible entry point for other possible carcinogens. CagA enters the cell through T4SS and is phosphorylated by Src, causing it to activate SHP-2, resulting in cellular morphological changes. Additionally, CagA can stabilize β-catenin, resulting in increased cell proliferation. Outer membrane protein BabA binds to human Lewis (b) surface epitopes [59,60], thus increasing specific proinflammatory cytokines like CXCL1 [61], that might boost cell proliferation [62]. Outer membrane protein OipA binds to EGFR, leading to activated PI3K-AKT signalling and increased β-catenin.

**Figure 3 pathogens-10-01321-f003:**
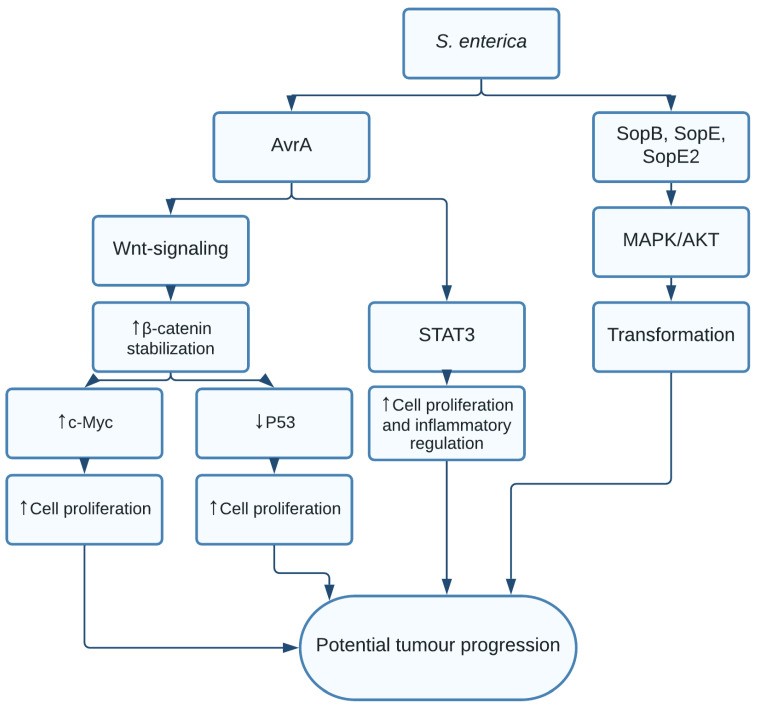
Oncogenic pathways activated by *Salmonella enterica:*
*Salmonella* protein AvrA activates Wnt-signalling, stabilizing β-catenin, leading to an increased expression of c-Myc and decreased expression of p53 resulting in increased cell proliferation. AvrA also activates STAT3-signaling, further increasing cell proliferation and inflammatory regulation. *Salmonella* proteins SopB, SopE and SopE2, injected through type III secretion system activate MAPK/AKT pathway provoking cell transformation and carcinoma.

**Figure 4 pathogens-10-01321-f004:**
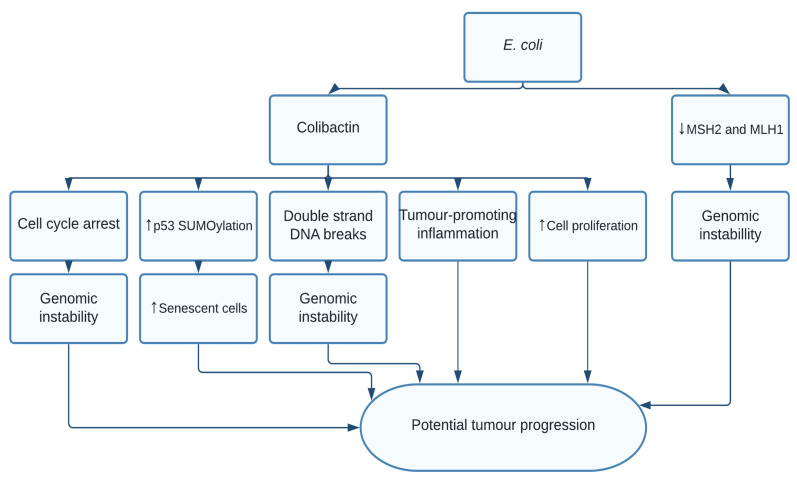
*E. coli*, through colibactin, caused cell cycle arrest and double-stranded DNA breaks resulting in genomic instability. Additionally, colibactin was found to increase cell proliferation and induce tumour-promoting inflammation. Colibactin also resulted in increased p53 SUMOylation resulting in emergence of senescent cells. Finally, *E. coli* downregulated mismatch repair proteins MSH2 and MLH1 leading to further genomic instability.

**Figure 5 pathogens-10-01321-f005:**
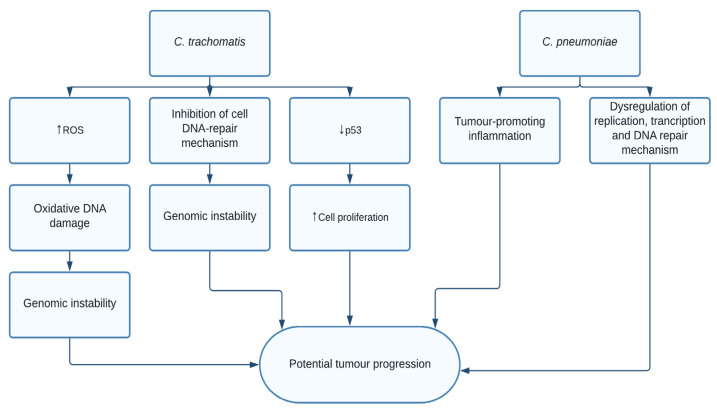
*C. trachomatis* downregulates the tumour suppressor P53 leading to increased cell proliferation, while also increasing levels of reactive oxygen species, causing oxidative DNA damage, which together with inhibition of cell DNA repair mechanisms causes genomic instability. *C. pneumoniae*, through unknown mechanisms, leads to inflammation and dysregulation of replication, transcription, and DNA repair mechanisms.

**Figure 6 pathogens-10-01321-f006:**
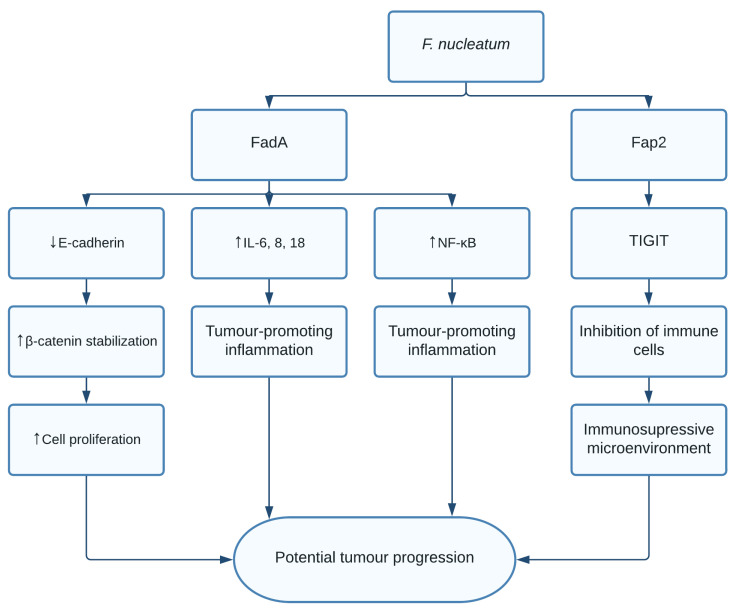
Pathways activated by *F. nucleatum*. FadA was found to inhibit E-cadherin, activating β-catenin and increasing cell proliferation. Additionally, increased levels of NF-κB and proinflammatory interleukins IL-6, IL-8, and IL-18 resulted in a possible tumour promoting inflammation. Fap2 showed interactions with the TIGIT immune receptor, causing inhibition of immune cells, and thereby creating an immunosuppressive microenvironment.

**Figure 7 pathogens-10-01321-f007:**
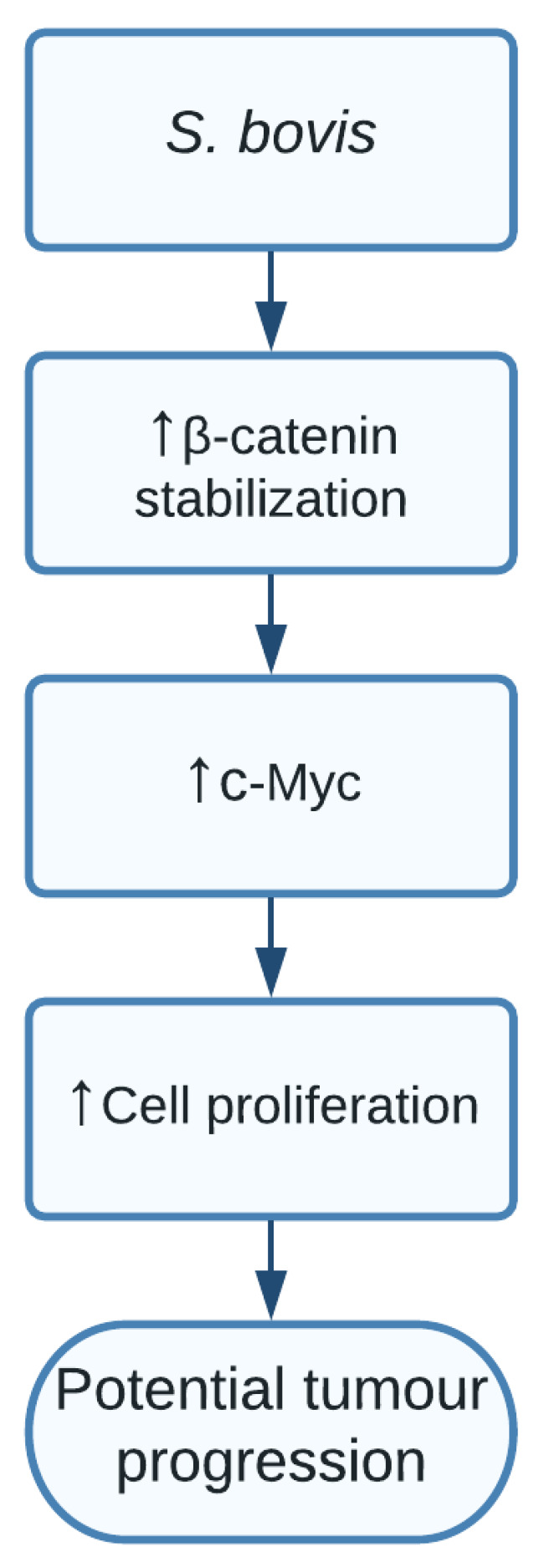
*S. bovis*, through an unknown pathway, increased levels of β-catenin and c-Myc, resulting in increased cell proliferation.

**Figure 8 pathogens-10-01321-f008:**
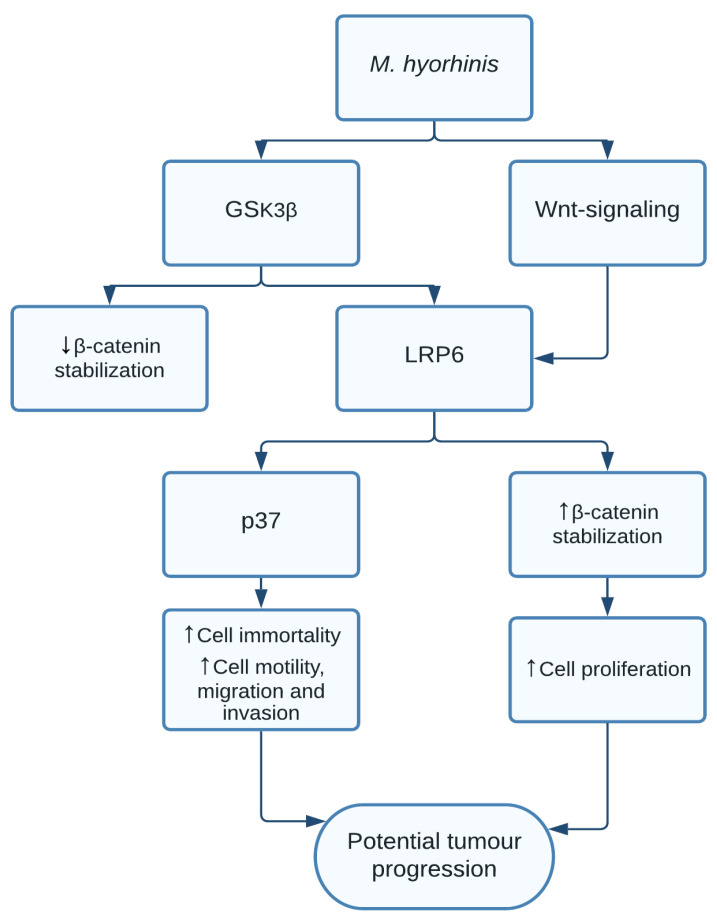
*M. hyorhinis* increases β-catenin through GSK3β, Wnt-signalling, and activation of LRP6. This resulted in increased cell proliferation. Without LRP6 activation, GSK3β had an inhibitory effect on β-catenin. LRP6 was additionally able to interact with p37, leading to increased levels of cell immortality, cell motility, migration, and invasion.

**Figure 9 pathogens-10-01321-f009:**
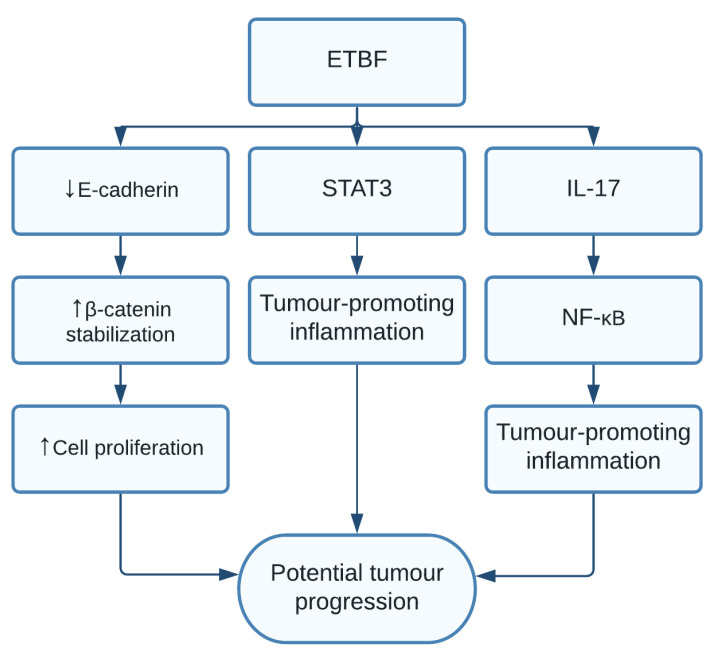
*B. fragilis* (ETBF) cleaves E-cadherin, leading to activated β-catenin levels, resulting in increased cell proliferation. ETBF also creates tumour-promoting inflammation through upregulation of the STAT3 pathway, IL-17, and NF-κB.

**Figure 10 pathogens-10-01321-f010:**
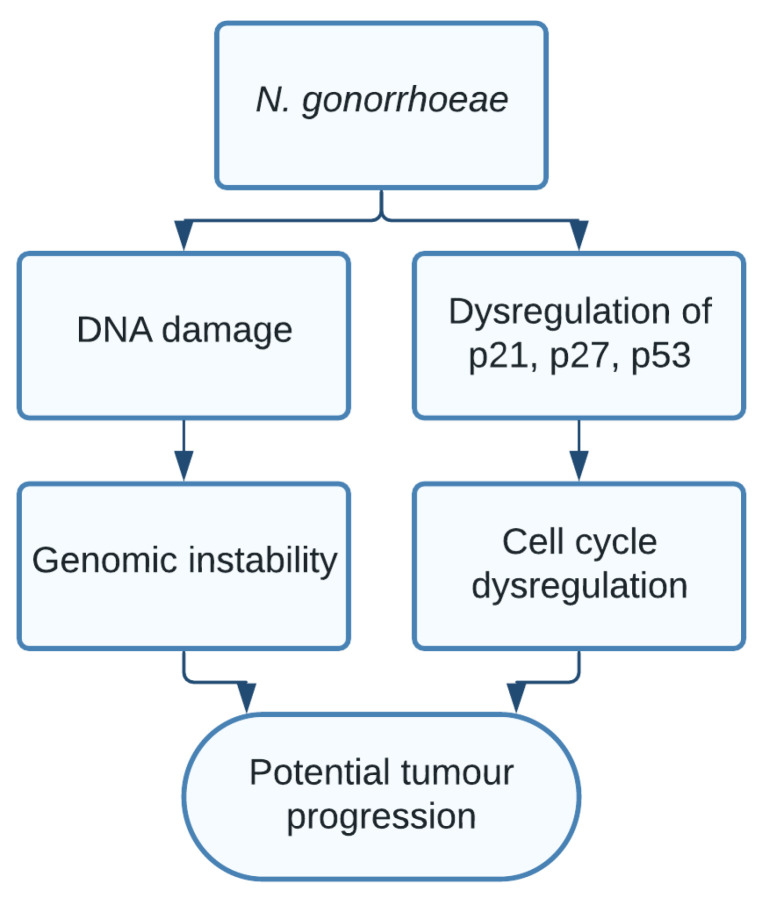
*N. gonorrhoeae*, through unclear mechanisms, caused DNA damage and increased the expression of p21 and p27 while decreasing the levels of p53 in non-tumour epithelial cells, resulting in genomic instability and cell cycle dysregulation.

**Figure 11 pathogens-10-01321-f011:**
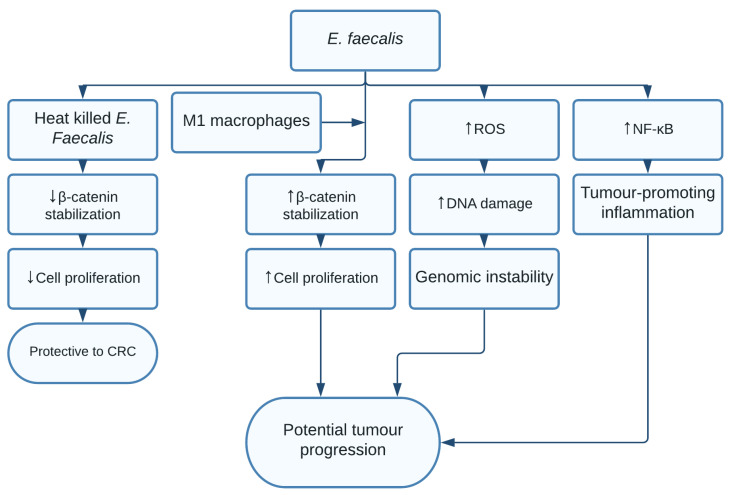
*E. faecalis* increased intracellular levels of ROS, leading to DNA instability and reduced DNA repair response resulting in genomic instability. E. Faecalis increased NF-κB levels, thereby causing a tumour promoting inflammation. When cocultured with M1 macrophages, β-catenin levels increased, leading to increased cell proliferation. Coculturing with heat-killed E. faecalis EC-12 strain, β-catenin was reduced, showing a possible protective role in CRC.

**Figure 12 pathogens-10-01321-f012:**
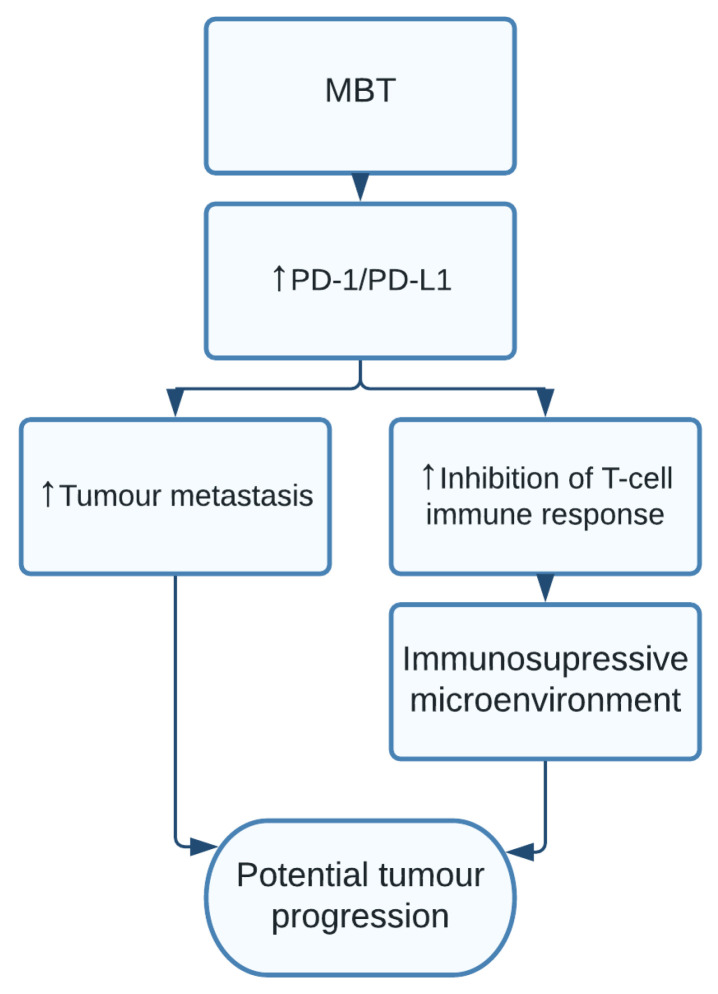
*Mycobacterium tuberculosis* (MBT) increased levels of PD-1/PD-L1 resulting in increased tumour metastasis while also inhibiting T cell immune response causing an immunosuppressive microenvironment favourable for potential tumour progression.

**Table 1 pathogens-10-01321-t001:** Overview of bacteria associated with carcinoma.

Bacteria	Potential Hallmarks of Cancer and Other Associated Pathologies	Type of Carcinoma Associated
*Helicobacter pylori*	Increase proliferation signalling [1,2,3]	Gastric cancer [1,2,3]
*Escherichia coli*(*pks*^+ve^ strain)	Genomic instability (DNA damage), tumour-promoting inflammation, and increased cell proliferation [4,5,6]	Colorectal cancer [4,5,6]
*Streptococcus bovis*	Increase proliferation signalling	Colorectal cancer [7,8]
*Salmonella enterica* (serovar *Enteritidis* and *Typhi*)	Increased cell proliferation, transformation, and tumour-promoting inflammation [9,10,11]	Colorectal cancer and gallbladder carcinoma [9,10,11]
*Fusobacterium nucleatum*	Increased cell proliferation, avoid immune destruction, and tumour-promoting inflammation [12,13]	Colorectal cancer (potentially oral, head, neck, oesophageal, cervical, and gastric cancer tissues) [12,13]
Enterotoxigenic *Bacteroid**es fragilis* (ETBF)	Increased cell proliferation and tumour-promoting inflammation [14,15,16]	Colorectal cancer [14,15,16]
*Enterococcus faecalis*	Genomic instability, tumour-promoting inflammation, and increased cell proliferation [17]	Colorectal cancer [17]
*Clostridium septicum*	Myonecrosis [18,19]	Potential involvement in colorectal cancer [19]
*Neisseria gonorrhoeae*	DNA damage [20], Genomic instability	Prostate cancer [21,22] and bladder cancer [23]
*Mycobacterium tuberculosis*	Tumour metastasis and avoiding immune destruction [24,25]	Lung cancer [24,25]
*Chlamydia trachomatis* and *Chlamydia pneumoniae*	Increased cell proliferation, tumour-promoting [26,27,28] inflammation [29], genomic instability [30], and evasion of apoptosis [31], evade immune system [32]	Cervical cancer [26,27,28] and lung cancer
*Mycoplasma hyorhinis*	Cell immortality, invasion, and increased cell proliferation [33,34]	Prostate cancer, gastric carcinoma, oesophageal cancer, lung cancer, and breast cancer [33,34]

## Data Availability

Not applicable.

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
