# Peer review of "Bacteria–Cancer Interface: Awaiting the Perfect Storm"

_pathogens, 2021, doi:10.3390/pathogens10101321_

Round 1

Reviewer 1 Report

Dear authors, 

the paper covers a quite interesting topic. I suggest that the manuscript will be extensively revised since there are many inaccuracies in the text. The major issue is that the first parts of this review covering H. pylori, Salmonella and Chlamydia as well as Introduction are well-described, though lucking some important discussion of recent data, which I indicated below. However, the part with E. coli is better to extend since this is another major (and well-studied) pathogen that might be associated with cancer. The other parts also require an extension, especially with the discussion of the recent literature, for instance, the parts with Clostridium and Mycobacterium, which are, in general, short in the discussion. The detailed comments and suggestions are below.

line14: not clear with "leading to activate invasion"

line17: correct "THEIR virulenCE factors"

line23: probably should write "virusES"

line32: "know cancer-associated bacterium" is also not clear

table 1: the table will benefit from giving appropriate references in front of each cancer type. For instance, "Gastric cancer [9, 10]" and so on. Readers could directly see the original paper.

line53: "has recently been associated" does not fit to VacA-induced gastric inflammation, it is already known for over 20 years...

line54: no need to capitalize "inflammation"

line57: some abbreviations like PI3K/AKT or GSK3ß are not defined, while others like CagA, SHP-2 are defined. I suggest keeping consistency.

line63: more common way of presenting Glu-Pro-Ile-Tyr-Ala is the EPIYA phosphorylation motif

line65: not clear with "leading of neoplasm"

line66: "which potential activates" is also not clear

line84: H. pylori should be in italics

figure 2: the resolution of the figure (here and after) should be increased. I think it's enough to abbreviate CagA and VacA in the main text. H. pylori - in italics.

line95: In respect to the H. pylori part of the review, I, unfortunately, did not see the discussion of recent advances in the field. Most of the discussed works are about 10-15 years old while there are some important recent data not included in this review. Specifically, cortactin is a well-known cellular target for this bacterium: https://doi.org/10.3390/cancers12010159. Moreover, it has been recently suggested as a host protein targeted by H. pylori in a CagA-dependent way, which likely contributes to gastric carcinogenesis. 

line102: "had been shown" please check the tense.

line105: it is better to abbreviate "type IV secretion system" as T4SS in line 61, and here give T3SS

line107: Interferon-gamma should be lowercased. Also, keep consistency with presenting Greek letters, before GSK3ß was written with Greek letter

lines112-113: STAT3 is abbreviated here while appearing for the first time in the previous part with H. pylori.

line114: check the reference 33 number. Also, "several studies" with one reference?

line123: MAPK is also abbreviated here while appearing for the first time in the previous part with H. pylori

line126: here TTSS is redundantly abbreviated.

line132: c-MYC differs from writing in line122 (c-Myc) or in Figure 3 (C-myc). Check consistency here and after

line137: I don't think there should be a shortened variant of E. coli in the subtitle (also in the following titles)

line138: please explain the meaning of "aero-anaerobic"

line140: colo- rectal? a means?

Figure 4: E. coli should be in italics

line187: Chlamydia trachomatis shortened to C. trachomatis in line 162 became even shorter (Ctr)...

line196: Single-strand should be in lowercase

lines209-210: check grammar

Figure 6 (here and after): species name should be in italics

Author Response

Dear reviewer

Please see the revision attached

Reviewer 2 Report

It would be better if this review could have a central theme that can make each section more cohesive. 

Other issues:

  • Line 12-14, What does this sentence "When healthy cells.... metastasis" means? There is a grammatical error
  • Line 1, "in this study" should be re-written as "in this review"
  • Line 22, add "leading to these genetic mutation" after "external factors"
  • Line 41, "have established" should be "has established"
  • Line 42, "a highest" should be "the highest"
  • Line 58, beta symbol need to be in correct font.
  • Figures 2-12 are better to be drawn in the format of graphics of bacteria, the host tissues and signaling pathways, visualising the interactions
  • Line 80-81, please explain how and what cytokines can cause increased cell proliferation
  • Line 99, add "of" after "increased cancer risk". and "respectively" should be put at the end of the sentence.
  • Line 102, replace "ending up in" with "and eventually cancer"
  • Line 102 to 104 (Two Salmonella.....), please correct the grammatical errors.
  • Line 105-106, (the second effector protein ...), please rearrange the phrase to make the sentence more logical.
  • Line 113, please add "," after the 2nd "and"
  • Line 117, what does it mean by "thus emerging"?
  • Line 145, "double-strand" should be replaced by "double-stranded"
  • Line 163-165, please rearrange the sentence starting with the Subspecies as the subject. It is because not all bacteria in this family would cause these infections.
  • Line 176, What does "changing of replication" means?
  • Line 211, "detection with" should be "detection by"
  • Line 212, "16S rDNA" should be "16S rRNA"
  •  Line 221, "paths" should be "pathways"
  • Line 222-223, the sentence need to be rewritten 
  • Line 263, please delete "or causal" 
  • Line 279, replace "divided" with "categorise"
  • Line 283, delete "to"
  • Line 285, please rephrase the sentence
  • Line 288, replace "to be" with "is"
  • line 306, delete "to", replace "and" with ", ", replace "increasing" with "increased"
  • Line 306-309, sentence is too long. Please divide it into multiple cohesive sentences
  • line 309, It should be figure 10
  • Line 339-340, what does this sentence mean?
  • line 354, delete "to"
  • Line 355-357, Please start with a new sentence
  • Line 367, "only recently have studies focused on bacterial factors that directly predispose infected tissue towards carcinoma." should be rewritten as "studies focused on bacterial factors that directly predispose infected tissue towards carcinoma have only recently been reported"
  • Line 368-368, What does this mean?
  • Line 371, what does  "If a bacteria is causal for cancer, why isn’t the cancer seen more frequently?" mean?  "a bacteria is" should be "bacteria are"
  • Line 373, what does "much is still not fully understood." mean?
  • Line 373-375, What does "memory" mean?
  • Line 378- 380, the sentences need to be re-written
  • LIne 390, what does this sentence mean?
  • Line 391-393, What does this sentence mean?
  • In the table and figure 1, please write the specific name of the bacteria that associate with harmful effect. E.g write ETBF rather than Bacteriodes fragilis as not all B. fragilis are harmful. Another example is E. Coli. Not all E. Coli associate with CRC

Author Response

Dear reviewer

Please see the revision attached.

Round 2

Reviewer 1 Report

The authors have addressed most of my commentaries. However, the discussion of recent advances in pathogen-induced cancers, as I requested in my first report, should still be extended. I provide the most relevant examples, that need to be discussed.

Helicobacter pylori

Line 72. The point with cortactin is that its overexpression is a major contribution to gastric (and many other types) carcinogenesis: https://www.wjgnet.com/1007-9327/full/v20/i12/3287.htm or more recent: https://link.springer.com/article/10.1007/s10238-021-00752-6 . So, recently H. pylori has been shown to upregulate cortactin expression  https://onlinelibrary.wiley.com/doi/10.1111/cmi.13376 that may at least partially explain the mechanism of the H. pylori-driven carcinogenic process. You could add cortactin in Figure 2, downstream of either CagA or pCagA (the cortactin upregulation by CagA is phospho-independent).

Escherichia coli

Add discussion in E. coli bacterial impact on bladder cancer cells through epithelial-mesenchymal transition, cancer stem cells and metabolic reprogramming: https://www.nature.com/articles/s41598-020-74390-5 

Fusobacterium nucleatum

There are at least two recent works extending our understanding of how F. nucleatum promotes carcinogenesis. Firstly, F. nucleatum infection may stimulate tumour cells to generate miR-1246/92b-3p/27a-3p-rich and CXCL16/RhoA/IL-8 exosomes that are delivered to uninfected cells to promote prometastatic behaviours: https://gut.bmj.com/content/70/8/1507.abstract . Another research shows that F. nucleatum can promote the development of colorectal cancer by activating a cytochrome P450/epoxyoctadecenoic acid axis via TLR4/Keap1/NRF2 signaling: https://cancerres.aacrjournals.org/content/81/17/4485.short 

Minors.

Check the spelling of "AVrA" in Figure 3.

line 122. Check spelling in "gall bladder"

line 150. Check speling in "E.coli "

Author Response

We thank the reviewer for pointing out the major studies to be extended and discussed in this review. We have made all the changes as suggested by the reviewer.

Point to point response.

Helicobacter pylori

Line 72. The point with cortactin is that its overexpression is a major contribution to gastric (and many other types) carcinogenesis: https://www.wjgnet.com/1007-9327/full/v20/i12/3287.htm or more recent: https://link.springer.com/article/10.1007/s10238-021-00752-6 . So, recently H. pylori has been shown to upregulate cortactin expression  https://onlinelibrary.wiley.com/doi/10.1111/cmi.13376 that may at least partially explain the mechanism of the H. pylori-driven carcinogenic process. You could add cortactin in Figure 2, downstream of either CagA or pCagA (the cortactin upregulation by CagA is phospho-independent).

We have incorporated the above said important cortactin link and references into the manuscript (line 82-85) as suggested by the reviewer. We have also made the corresponding changes in the figure 2.

Escherichia coli

Add discussion in E. coli bacterial impact on bladder cancer cells through epithelial-mesenchymal transition, cancer stem cells and metabolic reprogramming: https://www.nature.com/articles/s41598-020-74390-5 

We have incorporated the above study in the review (line 178-180).

Fusobacterium nucleatum

There are at least two recent works extending our understanding of how F. nucleatum promotes carcinogenesis. Firstly, F. nucleatum infection may stimulate tumour cells to generate miR-1246/92b-3p/27a-3p-rich and CXCL16/RhoA/IL-8 exosomes that are delivered to uninfected cells to promote prometastatic behaviours: https://gut.bmj.com/content/70/8/1507.abstract . Another research shows that F. nucleatum can promote the development of colorectal cancer by activating a cytochrome P450/epoxyoctadecenoic acid axis via TLR4/Keap1/NRF2 signaling: https://cancerres.aacrjournals.org/content/81/17/4485.short 

We have incorporated the above study in the review (line 270-275).

Minors.

Check the spelling of "AVrA" in Figure 3.

The following correction is made as pointed by the reviewer

line 122. Check spelling in "gall bladder"

The following correction is made as pointed by the reviewer

line 150. Check spelling in "E.coli "

E. Coli is italicized and the spelling is corrected throughout the manuscript.

Reviewer 2 Report

Thank you for the amendments. However, some minor issues could be resolved:

  1. Line 235, replace "cause" to "promote" cell proliferation
  2. Line 274, "Whether the found associations of these bacteria are causal or casual is yet unclear." does not make any sense. Please use another way to express this idea.

  3. Line 378-379, Please use another way to write this sentence "This also highlights if the bacterial association is a causal or casual factor for malignancies. "
  4. In the table, Please specify which strain of E. Coli. It is because not all E. Coli have these harmful effects. 

Author Response

We thank the reviewer for pointing out the minor revision.

Point to point response for the revision

1. Line 235, replace "cause" to "promote" cell proliferation

The above said change is incorporated in the manuscript.

2. Line 274, "Whether the found associations of these bacteria are causal or casual is yet unclear." does not make any sense. Please use another way to express this idea.

The suggested change is incorporated in the manuscript. We have rewritten the sentence as “whether these bacteria can directly lead to malignancies is yet unclear”.

3. Line 378-379, Please use another way to write this sentence "This also highlights if the bacterial association is a causal or casual factor for malignancies."

We have rewritten the sentence as “The bacterial effectors that can inflict mutations and result in malignancies in the infected tissue is yet to be discovered”.

4. In the table, Please specify which strain of E. Coli. It is because not all E. Coli have these harmful effects. 

We have specified the carcinogenic strain of E.Coli (pks +ve) in the table.
